# Synthesis of Novel Tricyclic N-Acylhydrazones as Tubulin Polymerization Inhibitors

**DOI:** 10.3390/ijms26189212

**Published:** 2025-09-20

**Authors:** Paola Corona, Michele Lai, Battistina Asproni, Giulia Sciandrone, Ilenia Lupinu, Roberta Ibba, Sandra Piras, Antonio Carta, Gabriele Murineddu

**Affiliations:** 1Department of Medicine, Surgery and Pharmacy, University of Sassari, 07100 Sassari, Italy; asproni@uniss.it (B.A.); robertaibbaphd@gmail.com (R.I.); piras@uniss.it (S.P.); acarta@uniss.it (A.C.); 2Retrovirus Centre, Department of Translational Medicine and New Technologies in Medicine and Surgery, University of Pisa, 56127 Pisa, Italy; michele.lai@unipi.it; 3Department of Medical Biotechnologies, University of Siena, 53100 Siena, Italy; g.sciandrone@student.unisi.it; 4Department of Chemical, Physical, Mathematical and Natural Sciences, University of Sassari, 07100 Sassari, Italy; i.lupinu@phd.uniss.it

**Keywords:** 1,4-dihydroindeno[1,2-*b*]pyrrole-3-carbohydrazides, A375 cell line, antiproliferative and cytotoxic activity, β-tubulin inhibitors, molecular docking

## Abstract

A series of *N*-acylhydrazones bearing a 1,4-dihydroindeno[1,2-*b*]pyrrole ring, along with benzene and thiophene rings substituted with chlorine or methyl groups, was synthesized and evaluated for their antiproliferative and cytotoxic activity against the melanoma A375 cell line and to measure the inhibition of tubulin polymerization. Four compounds elicited interesting activity: derivatives, **1g** and **1h** showed a 25% slowdown of tubulin polymerization, whereas compounds **2c** and **2d** caused a slowdown of 40% and 60%, respectively. Molecular modelling results have confirmed that the most active *N*-acylhydrazones (**1g**, **1h**, **2c**, and **2d**) may act as tubulin polymerization inhibitors.

## 1. Introduction

Cancer is one of the world’s leading health challenges, impacting millions of lives around the globe. It is responsible for approximately one in six (16.8%) of all deaths worldwide and one in four (22.8%) deaths from cardiovascular and chronic respiratory diseases and diabetes. It causes one in three (30.3%) premature deaths from non-communicable diseases (NCDs) globally, and it is among the three leading causes of death in the 30–69 age group in 177 out of 183 countries [1]. It is characterized by uncontrolled cell growth, the ability to invade surrounding tissue, and the capacity to spread to distant organs [2,3]. Advances in treatments such as surgery, chemotherapy, radiotherapy, and immunotherapy have been made. However, cancer continues to pose significant therapeutic challenges due to its complexity and genetic diversity and due to the development of drug resistance [4].

Moreover, conventional treatments often have severe side effects, which highlights the need for more effective, targeted therapies. For these reasons, the search for new anticancer agents with improved therapeutic profiles is a critical area of research [5], focusing on compounds that selectively target cancer cells while minimizing harm to normal tissues. In this context, small-molecule drugs have emerged as promising candidates for improving cancer treatment outcomes, particularly those capable of modulating key pathways involved in tumor growth and survival [6,7].

As part of a research project investigating new potential anticancer agents, we synthesized and characterized a class of 1,3,4-oxadiazole derivatives [8,9,10]. Among these, derivative 2-(2-methyl-1,4-dihydroindeno[1,2-*b*]pyrrol-3-yl)-5-phenyl-1,3,4-oxadiazole (**I**) (Figure 1) showed strong anti-proliferative activity against various human tumor cell lines, with IC_50_ values of 0.05 μM on HeLa (cervical adenocarcinoma cell line) and 1.7 μM on MCF-7 (breast cancer cell line) [8].

Further studies on its possible targets and molecular mechanisms revealed that derivative **I** is a tubulin inhibitor. It acts by mainly interfering with the cell cycle of cancer cells. It also affects the FoxO signalling pathway and the apoptotic and p53 signalling pathways.

The *TUBA1A* and *TUBA4A* genes, which code for the chain of alpha-1A tubulin and alpha-4A tubulin, respectively, are its candidate targets. Moreover, molecular docking results showed that compound **I** interacts at the colchicine binding site on tubulin [9].

In light of these results, we selected 2-(2-methyl-1,4-dihydroindeno[1,2-*b*]pyrrol-3-yl)-5-phenyl-1,3,4-oxadiazole (**I**) as the lead compound, and with the aim of expanding the structure–activity relationship (SAR) studies on this class of compounds, we investigated the effects of the ring opening of 1,3,4-oxadiazole to obtain *N*-acylhydrazone derivatives (Figure 2).

*N*-acylhydrazone compounds are an important class of bioactive compounds in medicinal chemistry. Their biological properties are connected to the presence of the reactive azomethine pharmacophore [11]. They exhibit a variety of biological effects, including anti-cancer properties [12,13,14,15]. In this context, we wanted to test this class of compounds on melanoma, since its incidence has increased greatly over the last two decades [16]. Therefore, the melanoma cell line A375 was selected to evaluate the antiproliferative and cytotoxic activity of the novel synthesized compounds and to measure the inhibition of tubulin polymerization.

We describe the synthesis, chemical characterization, inhibition of tubulin polymerization and docking studies of two novel series of tricyclic *N*-acylhydrazones, **1a**–**j** and **2a**–**d**,**i**,**j** (Table 1), which differ in terms the presence of a methyl group on the imine carbon in series 2.

## 2. Results and Discussion

### 2.1. Synthesis of Carbohydrazides

Carbohydrazides **1a**–**j** and **2a**–**d**,**i**,**j** were prepared by the four-step synthetic route reported in Figure 1. The key intermediate (**6**), which is required for synthesising the desired compounds, was prepared as previously reported [7], starting with the treatment of the commercially available compound 2-bromo-2,3-dihydro-*1H*-inden-1-one (**3**) with ethyl acetate and NaH in anhydrous THF to give the keto-ester **4**. The ester **4** was then cyclized to methyl pyrrole ester **5** by microwave irradiation in a suspension of NH_4_OAc and SiO_2_ in toluene. Then, derivative **5** was refluxed with hydrazine, leading to hydrazide **6**, whose reaction with the appropriate carbonyl derivatives yielded the desired compounds **1a**–**j** and **2a**–**d**,**i**,**j**.

Due to the assembly of amide and imine functions, carbohydrazides **1a**–**j** and **2a**–**d**,**i**,**j** can exist as C=N double-bond stereoisomers (E/Z). According to Palla and coworkers [17], all compounds synthesized exist in solution as the E geometric isomer, which is less sterically hindered than the Z form. Unequivocal structures were confirmed by a two-dimensional NOE spectroscopy (NOESY) experiment due to the proximity of the N=CH and CONH atoms in the structures of carbohydrazides **1a**–**j** and to the proximity of N=CCH_3_ and CONH in the case of derivatives **2a**–**d**,**i**,**j**. The NOESY experiment was carried out on compounds **1a** and **2a**, representative of the two series of carbohydrazides synthesized (Appendix A). In carbohydrazide **1a**, the singlet at 8.346 *δ*, which corresponds to N=CH, gives rise to a correlation peak with the amide hydrogen at 10.675 *δ*. For phenylethylidene carbohydrazide (**2a**), however, the same correlation occurs between the amide hydrogen at 10.67 *δ* and the singlet at 2.399 *δ*, which corresponds to N=CCH_3_.

### 2.2. Biology

#### 2.2.1. Immunocytochemistry

A375 cells were treated with each compound at a concentration of 10 µM for 48 h. Cells were then stained for the nucleus to count the number of surviving cells after 48 h of treatment and were also stained for β-tubulin to assess the effect on microtubules. After staining, the cells were analysed using high-content confocal microscopy screening (Figure 3A). Figure 3B shows the microscopy images collected after staining for nuclei or tubulin; cells treated with compound **2d** were selected as representative for the most active compounds. The images allow for the comparison between the vehicle-treated cells and the compound-treated ones, showing the massive loss of tubulin in the cytoplasm and the reduction in nuclei count (Figure 3C), proving high cytotoxicity. As shown in Figure 3D, we observed an increase in the nucleus/cytoplasm ratio in the presence of compounds **1g**, **1h**, **2c**, and **2d,** indicating a reduction in cytoplasmic area and indirectly suggesting microtubule destabilization. As reported in both graphs in Figure 3C,D, the hit compound **I** exerted a very high cytotoxicity and effectively altered the tubulin in the cytoplasm of A375 cells, confirming our previous data [9].

From the immunocytochemistry assay, compounds **1g**, **1h**, **2c**, and **2d** were selected for the best activity, and the ratio nucleus/cytoplasm index was the best scored; therefore, they were used for further evaluation on two different melanoma cell lines.

Using the same method just described the four selected compounds and compound **I** as a control were used to treat SK-MEL-28 cells of metastatic malignant melanoma (Figure 4) and OMM1 cells of metastatic uveal melanoma, a rare but very aggressive melanoma that originates in the eye (Figure 5). The cells were treated for 48 h, and as represented in Figure 4, the newly synthesized compounds were less active than parental compound **I**. Both the effect on the microtubules and the cytotoxic effect (detected by the nucleus count) was more prominent when SK-MEL-28 cells were treated with the control (compound **I**) than when treated with compounds **1g**, **1h**, **2c**, and **2d**. Among all, **2d** and **1h** were slightly more active than the others and had a comparable activity. When administered to OMM1 cells for 48 h at the same 10 µM concentration, compound **2c** emerged as the most active of the series with an analogous cytotoxic effect to the control compound **I**, and the microtubule disruption activity was also comparable, as depicted in Figure 5.

#### 2.2.2. Tubulin Polymerization Assay

To validate the findings, we performed an in vitro tubulin polymerization assay. This assay measures fluorescence enhancement resulting from the incorporation of a fluorescence reporter into microtubules during polymerization and is useful for assessing the effects of compounds or proteins on tubulin polymerization (Figure 6A). Our results confirmed that **1g**, **1h**, **2c**, and **2d** slowed down tubulin polymerization by 25%, 25%, 40%, and 60%, respectively, as shown in Figure 6B.

### 2.3. Computational Assay

#### 2.3.1. Computational Evaluation of Ligand–Tubulin Interactions

To assess the potential binding modes of the synthesized compounds to the target protein tubulin, molecular docking simulations were performed using AutoDock Vina 1.1.2 [18].

The top-ranked binding poses revealed binding affinities ranging from −10.0 to −8.8 kcal/mol across the compound series. For comparison, colchicine was re-docked into its native binding pocket, showing an affinity of −10.7 kcal/mol (Table 2).

Multiple poses were generated for each compound. The best binding pose was selected based on the geometry of key interactions with critical residues, overall pose consistency, and the plausibility of hydrogen bonding and hydrophobic interactions, rather than merely on the lowest binding energy. The binding poses, the main interactions, and the images were evaluated using BIOVIA Discovery Studio Visualizer v25.1.0.24284, Dassault Systèmes, 2025 [19].

#### 2.3.2. Binding Mode Analysis of Compound **2c**

Among the four most active compounds identified in our biological screening (**1g**, **1h**, **2c**, and **2d**), compound **2c** exhibited the most overall favourable docking score and was selected for analysis. Its predicted binding mode within the colchicine binding site is shown in Figure 7.

Through visual inspection of the top-ranked poses, the selected conformation of **2c** displayed a consistent interaction pattern involving both hydrogen bonding (Asn258) and Pi–alkyl contacts with Leu255, Cys241, and Ala316. Despite some broader distances (>4.5 Å), the overall fit and spatial arrangement are in line with those seen for colchicine-like ligands, and the pose showed high reproducibility across compounds.

#### 2.3.3. Structural Superimposition with Colchicine

The colchicine binding site is a well-established and therapeutically relevant target for the development of new anticancer agents due to its critical role in inhibiting microtubule polymerization, thereby disrupting cell division and inducing apoptosis in rapidly proliferating cancer cells [20].

To further support the plausibility of the predicted binding mode, a superimposition with colchicine was performed. As illustrated in Figure 8, the docked compound occupies a similar region within the binding pocket, partially overlapping with colchicine and mimicking key interaction motifs. This structural alignment supports a shared binding mechanism with potential tubulin inhibition.

The predicted binding of the studied compounds at the colchicine site, similarly to previously reported derivatives [10], suggests and supports that they may exert cytotoxic activity by destabilizing microtubules.

#### 2.3.4. Binding Patterns of the Most Active Ligands

A broader docking analysis was conducted for the four most active compounds (**1g**, **1h**, **2c**, and **2d**). Their binding conformations and interaction patterns within the colchicine pocket are displayed in Figure 9.

All compounds formed a hydrogen bond with Asn258, highlighting its feasible critical role. Among them, compound **2c** established the most extensive interaction network, forming a hydrogen bond with Asn258, a Pi–donor hydrogen bond with Cys101, and multiple Pi–alkyl interactions with Lys352, Ala250, Leu248, Ala316, Leu255, and Cys241.

Compound **2d** maintained the Asn258 hydrogen bond and formed a Pi–cation interaction with Lys254 but displayed fewer hydrophobic contacts. Compound **1g** showed an intermediate profile with Pi–alkyl contacts involving Leu248, Ala250, and Lys352, while **1h** formed a Pi–cation interaction with Lys254 and retained the key Asn258 hydrogen bond.

Binding distances ranged from 2.38 to 5.50 Å, with most interactions falling within favourable ranges for stable protein–ligand binding. These differences in interaction networks may account for the small variations in docking scores and potential activity.

#### 2.3.5. Methodological Considerations

Molecular docking is a powerful and accessible method for early-stage drug discovery, nonetheless it is important to acknowledge its limitations. Considering that the differences in docking scores obtained in this study are limited and generally below 1 kcal/mol and given the inherent sensitivity limitations of current scoring functions, reliable discrimination of biological activity based on such a narrow range is not feasible. This limitation is well acknowledged in the literature, where docking scores rarely provide meaningful quantitative correlations when score variations are minimal or when biological data are limited [21,22]. Therefore, in this study, the docking results have been interpreted qualitatively to assess the binding potential of the analysed compounds toward the target protein tubulin, in line with common practice in the field. Furthermore, current docking algorithms are constrained by the accuracy of input molecular structures, approximations in conformational sampling, and the limited predictive power of scoring functions [23,24]. Therefore, experimental validation through biochemical and cellular assays remains essential to confirm computational predictions.

## 3. Methods and Materials

### 3.1. Chemistry

#### 3.1.1. General Methods

Melting points were determined with a Köpfler apparatus and are uncorrected.

Thin-Layer Chromatography (TLC) was performed on Poligram^®^ (Kieselgel 60 F254, Merck^®^) SIL N-HR/HV_254_ silica plates (0.2 mm). Compounds were purified by flash chromatography (FC) automatically on a Flash-master (Biotage^®^, Uppsala, Sweden) with pre-packed Biotage^®^ SNAP silica gel cartridges or manually on silica gel (Kieselgel 60, 0.040−0.063 mm, Merck^®^, Darmstad, Germany).

^1^H and ^13^C NMR spectra were recorded at room temperature with a Bruker^®^ (Fallanden, Switzerland) AVANCE III Nanoboy 400 MHz spectrophotometer using TMS as an internal standard. Spectra were acquired using deuterated dimethyl sulfoxide (DMSO-d_6_) as solvents. Chemical shifts are expressed in *δ* units and the coupling constants, *J*, in Hz. Multiplicities are indicated as s (singlet), br s (broad singlet), d (doublet), dd (double doublet), t (triplet), and m (multiplet). (Appendix A). Spectroscopical data are consistent with structures.

IR spectra were recorded in a KBr tablet with a Jasco^®^ (Cremella, Italy) FT/IR460 plus spectrophotometer, and absorbance is indicated as a wave number (ν, cm^−1^).

All reactions involving air or moisture-sensitive compounds were performed under a nitrogen or argon atmosphere.

LC/MS analyses were performed on an Agilent (Santa Clara, CA, USA) 1100 LC/MSD system consisting of a single quadrupole detector (SQD) mass spectrometer (MS) equipped with an electrospray ionization (ESI) interface and a photodiode array (PDA) detector, with a range of 120−550 nm. ESI in the positive mode was applied. Mobile phases were as follows: (A) MeOH in H_2_O (8:2). Analyses were performed at a flow rate of 0.9 mL/min and temperature of 350 °C. The purity of all final compounds was determined by elemental analysis on a PerkinElmer^®^ (Waltham, MA, USA) 240-B analyser (C, H, and N). All the final compounds were found to be >95% when analysed.

Reagents and solvents were purchased from Merck, Alfa Aesar (GB), and Acros Organics (Belgium) and were used without further purification or drying.

#### 3.1.2. General Procedure for the Synthesis of Carbohydrazides **1a**–**j** and **2a**–**d**,**i**,**j**

The appropriate carbonyl derivatives (1 eq) were added to a suspension of 2-methyl-1,4-dihydroindeno[1,2-*b*]pyrrole-3-carbohydrazide (**6**) (0.100 g, 0.32 mmol, 1 eq) in EtOH (3.6 mL). To catalyse the reaction, for **1a**–**j**, two drops of concentrated HCl 6M were also added. The mixture was stirred at room temperature for 0.5–48 h. The resulting precipitate was filtered, washed with water, and air-dried to produce the desired products **1a**–**j** and **2a**–**d**,**i**,**j**.

##### (*E*)-*N*′-(Benzylidene)-2-methyl-1,4-dihydroindeno[1,2-*b*]pyrrole-3-carbohydrazide (**1a**)

The title compound was prepared from **6** and benzaldehyde using the general procedure after stirring for 0.5 h to afford **1a** as a cream solid (0.11 g, 78%), mp = 235–236 °C. IR (cm^−1^): 1628.40 (C=O), 3255.86 (NH), 3050.89 (NH). ^1^H NMR (400 MHz, DMSO-d_6_) *δ*: 2.55 (s, 3H), 3.73 (s, 2H), 7.05 (t, 1H, *J* = 7.2 Hz), 7.25 (t, 1H, *J* = 7.2 Hz), 7.35 (d, 1H, *J* = 7.6 Hz), 7.42–7.46 (m, 4H), 7.67–7.69 (m, 2H), 8.35 (s, 1H), 10.67; ^13^C NMR (100 MHz, DMSO-d_6_) *δ*: 13.34 (CH_3_), 31.21 (CH_2_), 110.46 (C), 115.79 (CH), 122.73 (CH), 124.98 (CH), 126.39 (CH), 126.70 (CH × 2), 128.80 (CH × 3), 129.56 (CH), 134.67 (C × 2), 135.03 (C × 2), 137.05 (C), 145.62 (C × 2). MS (ESI): C_20_H_17_N_3_O requires m/z 315.38, found 316.38 [M+H]^+^. Anal. calcd for C_20_H_17_N_3_O: C, 76.17; H, 5.43; N, 13.32. Found: C, 75.94; H, 5.41; N, 13.29.

##### (*E*)-*N*′-(4-Chlorobenzylidene)-2-methyl-1,4-dihydroindeno[1,2-*b*]pyrrole-3-carbohydrazide (**1b**)

The title compound was prepared from **6** and 4-chlorobenzaldehyde using the general procedure after stirring for 1.5 h to afford **1b** as a light brown solid (0.10 g, 69%), mp = 202–203 °C. IR (cm^−1^): 1632.20 (C=O), 3258.70 (NH), 3049.80 (NH). ^1^H NMR (400 MHz, DMSO-d_6_) *δ*: 2.54 (s, 3H), 3.71 (s, 2H), 7.05 (t, 1H, *J* = 7.6 Hz), 7.25 (t, 1H, *J* = 8.0 Hz), 7.34 (d, 1H, *J* = 7.6 Hz), 7.43 (d, 1H, *J* = 7.6 Hz), 7.51 (d, 2H, *J* = 8.4 Hz), 7.70 (d, 2H, *J* = 7.6 Hz), 8.34 (s, 1H), 10.73 (br s, 1H, NH exch. with D_2_O), 11.73 (s, 1H, NH exch. with D_2_O); ^13^C NMR (100 MHz, DMSO-d_6_) *δ*: 13.34 (CH_3_), 31.20 (CH_2_), 110.42 (C), 115.80 (CH), 122.73 (CH), 124.98 (CH), 126.39 (CH), 128.29 (CH), 128.71 (CH × 2), 128.89 (CH × 2), 129.61 (C), 133.67 (C), 133.92 (C), 134.72 (C), 135.03 (C), 137.14 (C), 137.76 (C), 145.60 (C). MS (ESI): C_20_H_16_ClN_3_O requires *m*/*z* 349.82, found 350.82 [M+H]^+^. Anal. calcd for C_20_H_16_ClN_3_O: C, 68.67; H, 4.61; N, 12.01. Found: C, 68.53; H, 4.60; N, 11.99.

##### (*E*)-*N*′-(3-Chlorobenzylidene)-2-methyl-1,4-dihydroindeno[1,2-*b*]pyrrole-3-carbohydrazide (**1c**)

The title compound was prepared from **6** and 3-chlorobenzaldehyde using the general procedure after stirring for 0.5 h to afford **1c** as a cream solid (0.12 g, 81%), mp = 195–196 °C. IR (cm^−1^): 1636.50 (C=O), 3255.70 (NH), 3047.30 (NH). ^1^H NMR (400 MHz, DMSO-d_6_) *δ*: 2.54 (s, 3H), 3.72 (s, 2H), 7.05 (t, 1H, *J* = 8,0 Hz), 7.25 (t, 1H, *J* = 8.0 Hz), 7.34 (d, 1H, *J* = 7.6 Hz), 7.41–7.50 (m, 3H), 7.62–7.64 (m, 1H), 7.74 (s, 1H), 8.32 (s, 1H), 10.81 (br s, 1H, NH exch. with D_2_O), 11.74 (s, 1H, NH exch. with D_2_O); ^13^C NMR (100 MHz, DMSO-d_6_) *δ*: 13.83 (CH_3_), 31.80 (CH_2_), 110.83 (C), 116.29 (CH), 123.23 (CH), 125.44 (CH), 125.95 (CH), 126.29 (CH × 2), 126.88 (CH), 129.63 (CH), 131.20 (CH), 134.08 (C), 135.24 (C), 135.49 (C × 2), 137.47 (C × 2), 137.72 (C), 146.08 (C). MS (ESI): C_20_H_16_ClN_3_O requires *m*/*z* 349.82, found 350.82 [M+H]^+^. Anal. calcd for C_20_H_16_ClN_3_O: C, 68.67; H, 4.61; N, 12.01. Found: C, 68.46; H, 4.60; N, 11.97.

##### (*E*)-*N*′-(2-Chlorobenzylidene)-2-methyl-1,4-dihydroindeno[1,2-*b*]pyrrole-3-carbohydrazide (**1d**)

The title compound was prepared from **6** and 2-chlorobenzaldehyde using the general procedure after stirring for 1 h to afford **1d** as a cream solid (0.14 g, 92%), mp = 229–233 °C. IR (cm^−1^): 1638.30 (C=O), 3259.40 (NH), 3049.50 (NH). ^1^H NMR (400 MHz, DMSO-d_6_) *δ*: 2.54 (s, 3H), 3.73 (s, 2H), 7.05 (t, 1H, *J* = 8.0 Hz), 7.25 (t, 1H, *J* = 7.6 Hz), 7.34 (d, 1H, *J* = 7.6 Hz), 7.38–7.59 (m, 4H), 7.97 (s, 1H), 8.74 (s, 1H), 10.98 (br s, 1H, NH exch. with D_2_O), 11.74 (s, 1H, NH exch. with D_2_O); ^13^C NMR (100 MHz, DMSO-d_6_) *δ*: 13.37 (CH_3_), 31.19 (CH_2_), 110.33 (C), 115.80 (CH), 122.75 (CH), 124.99 (CH), 126.37 (CH), 127.52 (CH × 2), 129.88 (CH × 2), 130.93 (CH), 131.99 (C), 132.79 (C), 134.77 (C), 135.00 (C × 2), 137.33 (C), 145.65 (C × 2). MS (ESI): C_20_H_16_ClN_3_O requires *m*/*z* 349.82, found 350.82 [M+H]^+^. Anal. calcd for C_20_H_16_ClN_3_O: C, 66.67; H, 4.61; N, 12.01. Found: C, 66.07; H, 4.49; N, 11.87.

##### (*E*)-2-Methyl-*N*′-(2-methylbenzylidene)-1,4-dihydroindeno[1,2-*b*]pyrrole-3-carbohydrazide (**1e**)

The title compound was prepared from **6** and 2-methylbenzaldehyde using the general procedure after stirring for 0.5 h to afford **1e** as a cream solid (0.12 g, 87%), mp = 207–208 °C. IR (cm^−1^): 1645.70 (C=O), 3263.40 (NH), 3052.50 (NH). ^1^H NMR (400 MHz, DMSO-d_6_) *δ*: 2.45 (s, 3H), 2.55 (s, 3H), 3.74 (s, 2H), 7.05 (t, 1H, *J* = 7.6 Hz), 7.23–7.31 (m, 4H), 7.35 (d, 1H, *J* = 7.6 Hz), 7.44 (d, 1H, *J* = 7.6 Hz), 7.76–7.84 (m, 1H), 8.63 (s, 1H), 10.65 (s, 1H, NH exch. with D_2_O), 11.71 (s, 1H, NH exch. with D_2_O); ^13^C NMR (100 MHz, DMSO-d_6_) *δ*: 13.36 (CH_3_), 19.04 (CH_3_), 31.20 (CH_2_), 110.47 (C), 115.78 (CH), 122.72 (CH), 124.97 (CH), 126.08 (CH × 2), 126.38 (CH), 129.22 (CH), 130.80 (CH × 2), 132.66 (C × 2), 134.69 (C), 135.05 (C), 136.45 (C), 137.05 (C), 145.62 (C × 2). MS (ESI): C_21_H_19_N_3_O requires *m*/*z* 329.40, found 330.40 [M+H]^+^. Anal. calcd for C_21_H_19_N_3_O: C, 76.57; H, 5.81; N, 12.76. Found C, 76.34; H, 5.79; N, 12.72.

##### (*E*)-*N*′-(2,4-Dichlorobenzylidene)-2-methyl-1,4-dihydroindeno[1,2-*b*]pyrrole-3-carbohydrazide (**1f**)

The title compound was prepared from **6** and 2,4-dichlorobenzaldehyde using the general procedure after stirring for 2 h to afford **1f** as a cream solid (0.16 g, 92%), mp = 237–238 °C. IR (cm^−1^): 1646.90 (C=O), 3248.30 (NH), 3049.20 (NH). ^1^H NMR (400 MHz, DMSO-d_6_) *δ*: 2.54 (s, 3H), 3.73 (s, 2H), 7.05 (t, 1H, *J* = 8.4 Hz), 7.25 (t, 1H, *J* = 8.0 Hz), 7.35 (d, 1H, *J* = 7.6 Hz), 7.44 (d, 1H, *J* = 6.4 Hz), 7.51 (d, 1H, *J* = 8.4 Hz), 7.71 (d, 1H, *J*_m_ = 2.0 Hz), 7.97 (s, 1H), 8.70 (s, 1H), 11.03 (br s, 1H, NH exch. with D_2_O), 11.76 (s, 1H, NH exch. with D_2_O); ^13^C NMR (100 MHz, DMSO-d_6_) *δ*: 13.36 (CH_3_), 31.20 (CH_2_), 110.25 (C), 115.81 (CH), 122.78 (CH), 125.01 (CH), 126.38 (CH), 127.78 (CH), 127.94 (CH × 2), 129.32 (CH), 131.13 (C), 133.43 (C), 134.48 (C), 134.81 (C), 134.97 (C × 2), 137.43 (C), 145.63 (C × 2). MS (ESI): C_20_H_15_Cl_2_N_3_O requires *m*/*z* 384.26, found 394.26 [M+H]^+^. Anal. calcd for C_20_H_15_Cl_2_N_3_O: C, 62.51; H, 3.93; N, 10.94. Found: C, 62.32; H, 3.92; N, 10.91.

##### (*E*)-*N*′-(3,5-Dichlorobenzylidene)-2-methyl-1,4-dihydroindeno[1,2-*b*]pyrrole-3-carbohydrazide (**1g**)

The title compound was prepared from **6** and 3,5-dichlorobenzaldehyde using the general procedure after stirring for 0.5 h to afford **1g** as a yellow solid (0.16 g, 94%), mp = 183–184 °C. IR (cm^−1^): 1651.60 (C=O), 3252.40 (NH), 3055.20 (NH). ^1^H NMR (400 MHz, DMSO-d_6_) *δ*: 2.54 (s, 3H), 3.71 (s, 2H), 7.06 (t, 1H, *J* = 7.6 Hz), 7.25 (t, 1H, *J* = 7.6 Hz), 7.35 (d, 1H, *J* = 7.6 Hz), 7.43 (d, 1H, *J* = 7.2 Hz), 7.64 (s, 1H), 7.71 (s, 2H), 8.30 (s, 1H), 10.95 (br s, 1H, NH exch. with D_2_O), 11.78 (s, 1H, NH exch. with D_2_O); ^13^C NMR (100 MHz, DMSO-d_6_) *δ*: 13.35 (CH_3_), 31.18 (CH_2_), 110.24 (C), 115.84 (CH), 122.80 (CH), 124.92 (CH × 3), 124.97 (CH), 126.42 (CH), 128.50 (CH), 134.57 (C × 2), 134.83 (C), 134.96 (C × 2), 137.41 (C), 138.51 (C × 2), 145.57 (C). MS (ESI): C_20_H_15_Cl_2_N_3_O requires *m*/*z* 384.26, found 394.26 [M+H]^+^. Anal. calcd for C_20_H_15_Cl_2_N_3_O: C, 62.51; H, 3.93; N, 10.94. Found: C, 62.32; H, 3.91; N, 10.91.

##### (*E*)-*N*′-(2,6-Dichlorobenzylidene)-2-methyl-1,4-dihydroindeno[1,2-*b*]pyrrole-3-carbohydrazide (**1h**)

The title compound was prepared from **6** and 2,6-dichlorobenzaldehyde using the general procedure after stirring for 2 h to afford **1h** as a cream solid (0.15 g, 87%), mp > 250 °C. IR (cm^−1^): 1649.30 (C=O), 3250.50 (NH), 3052.10 (NH). ^1^H NMR (400 MHz, DMSO-d_6_) *δ*: 2.54 (s, 3H), 3.72 (s, 2H), 7.04 (t, 1H, *J* = 8.4 Hz), 7.24 (t, 1H, *J* = 8.0 Hz), 7.34 (d, 1H, *J* = 7.2 Hz), 7.41–7.46 (m, 2H), 7.57 (d, 2H, *J* = 8.0 Hz), 8.53 (s, 1H), 11.01 (br s, 1H, NH exch. with D_2_O), 11.74 (s, 1H, NH exch. with D_2_O); ^13^C NMR (100 MHz, DMSO-d_6_) *δ*: 13.47 (CH_3_), 31.20 (CH_2_), 110.12 (C), 115.79 (CH), 122.75 (CH), 124.95 (CH), 126.36 (CH), 129.03 (CH × 3), 130.84 (CH), 130.91 (C × 2), 133.87 (C × 2), 134.77 (C), 134.98 (C × 2), 137.56 (C), 145.63 (C). MS (ESI): C_20_H_15_Cl_2_N_3_O requires *m*/*z* 384.26, found 394.26 [M+H]^+^. Anal. calcd for C_20_H_15_Cl_2_N_3_O: C, 62.51; H, 3.93; N, 10.94. Found: C, 62.38; H, 3.85; N, 10.92.

##### (*E*)-2-Methyl-*N*′-(thiophen-2-ylmethylene)-1,4-dihydroindeno[1,2-*b*]pyrrole-3-carbohydrazide (**1i**)

The title compound was prepared from **6** and thiophene-2-carbaldehyde using the general procedure after stirring for 0.5 h to afford **1i** as a cream solid (0.03 g, 24%), mp = 246–247 °C. IR (cm^−1^): 1652.80 (C=O), 3254.10 (NH), 3055.70 (NH). ^1^H NMR (400 MHz, DMSO-d_6_) *δ*: 2.53 (s, 3H), 3.71 (s, 2H), 7.05 (t, 1H, *J* = 7.2 Hz), 7.12–7.14 (m, 1H), 7.25 (t, 1H, 7.2 Hz), 7.34 (d, 1H, *J* = 7.6 Hz), 7.39 (d, 1H, *J* = 3.6 Hz), 7.43 (d, 1H, *J* = 7.6 Hz), 7.62 (d, 1H, *J* = 4.4 Hz), 8.56 (s, 1H), 10.65 (br s, 1H, NH exch. with D_2_O), 11.70 (s, 1H, NH exch. with D_2_O); ^13^C NMR (100 MHz, DMSO-d_6_) *δ*: 13.35 (CH_3_), 31.20 (CH_2_), 110.43 (C), 115.77 (CH), 122.70 (CH), 124.95 (CH), 126.37 (CH), 127.75 (CH), 128.12 (CH), 129.90 (CH × 2), 134.48 (C), 135.04 (C × 2), 136.85 (C), 139.52 (C), 145.65 (C × 2). MS (ESI): C_18_H_15_N_3_OS requires *m*/*z* 321.40, found 322.40 [M+H]^+^. Anal. calcd for C_18_H_15_N_3_OS: C, 67.27; H, 4.70; N, 13.07. Found: C, 67.14; H, 4.69; N, 13.04.

##### (*E*)-2-Methyl-*N*′-(thiophen-3-ylmethylene)-1,4-dihydroindeno[1,2-*b*]pyrrole-3-carbohydrazide (**1j**)

The title compound was prepared from **6** and thiophene-3-carbaldehyde using the general procedure after stirring for 2 h to afford **1j** as a light brown solid (0.08 g, 52%), mp = 175–176 °C. IR (cm^−1^): 1642.90 (C=O), 3258.13 (NH), 3048.70 (NH). ^1^H NMR (400 MHz, DMSO-d_6_) *δ*: 2.53 (s, 3H), 3.70 (s, 2H), 7.04 (t, 1H, *J* = 8.0 Hz), 7.24 (t, 1H, *J* = 7.6 Hz), 7.34 (d, 1H, *J* = 7.6 Hz), 7.41–7.43 (m, 2H), 7.61–7.63 (m, 1H), 7.84 (d, 1H, *J* = 2.8 Hz), 8.37 (s, 1H), 10.64 (br s, 1H, NH exch. with D_2_O), 11.70 (s, 1H, NH exch. with D_2_O); ^13^C NMR (100 MHz, DMSO-d_6_) *δ*: 13.33 (CH_3_), 31.15 (CH_2_), 110.58 (C), 115.77 (CH), 122.68 (CH), 124.54 (CH), 124.97 (CH × 2), 126.38 (CH), 127.04 (CH), 127.51 (CH), 134.64 (C × 2), 135.07 (C), 136.87 (C), 137.85 (C), 145.62 (C × 2). MS (ESI): C_18_H_15_N_3_OS requires *m*/*z* 321.40, found 322.40 [M+H]^+^. Anal. calcd for C_18_H_15_N_3_OS: C, 67.27; H, 4.70; N, 13.07. Found: C, 67.07; H, 4.69; N, 13.03.

##### (*E*)-2-Methyl-*N*′-(1-phenylethylidene)-1,4-dihydroindeno[1,2-*b*]pyrrole-3-carbohydrazide (**2a**)

The title compound was prepared from **6** and acetophenone using the general procedure after stirring for 10 h to afford **2a** as a white solid (0.12 g, 83%), mp > 250 °C. IR (cm^−1^): 1655.71 (C=O), 3244.49 (NH), 3095.94 (NH). ^1^H NMR (400 MHz, DMSO-d_6_) *δ*: 2.40 (s, 3H), 2.58 (s, 3H), 3.73 (s, 2H), 7.05 (t, 1H, *J* = 8.0 Hz), 7.25 (t, 1H, *J* = 8.0 Hz), 7.35 (d, 1H, *J* = 7.6 Hz), 7.40–7.46 (m, 4H), 7.83 (d, 2H, *J* = 7.2 Hz), 9.63 (br s, 1H, NH exch. with D_2_O), 11.74 (s, 1H, NH exch. with D_2_O); ^13^C NMR (100 MHz, DMSO-d_6_) *δ*: 13.95 (CH_3_), 14.08 (CH_3_), 31.66 (CH_2_), 111.06 (C), 116.29 (CH), 123.23 (CH), 125.52 (CH), 126.61 (CH × 2), 126.92 (CH), 128.78 (CH × 2), 129.41 (CH), 135.16 (C), 135.53 (C × 2), 137.77 (C × 2), 138.86 (C × 2), 146.00 (C). MS (ESI): C_21_H_19_N_3_O requires *m*/*z* 329.40, found 330.40 [M+H]^+^. Anal. calcd for C_21_H_19_N_3_O: C, 76.57; H, 5.81; N, 12.76. Found: C, 76.42; H, 5.80; N, 12.73.

##### (*E*)-*N*′-(1-(4-Chlorophenyl)ethylidene)-2-methyl-1,4-dihydroindeno[1,2-*b*]pyrrole-3-carbohydrazide (**2b**)

The title compound was prepared from **6** and 1-(4-chlorophenyl)ethan-1-one using the general procedure after stirring for 10 h to afford **2b** as a cream solid (0.12 g, 77.5%), mp > 250 °C. IR (cm^−1^): 1657.32 (C=O), 3246.17 (NH), 3098.22 (NH). ^1^H NMR (400 MHz, DMSO-d_6_) *δ*: 2.39 (s, 3H), 2.58 (s, 3H), 3.72 (s, 2H), 7.06 (t, 1H, *J* = 7.2 Hz), 7.26 (t, 1H, *J* = 8.0 Hz), 7.35 (d, 1H, *J* = 7.2 Hz), 7.45 (d, 1H, *J* = 7.6 Hz), 7.49 (d, 2H, *J* = 8.4 Hz), 7.85 (d, 2H, *J* = 8.8 Hz), 9.68 (br s, 1H, NH exch. with D_2_O), 11.75 (s, 1H, NH exch. with D_2_O); ^13^C NMR (100 MHz, DMSO-d_6_) *δ*: 13.44 (CH_3_), 13.46 (CH_3_), 31.18 (CH_2_), 110.52 (C), 115.82 (CH), 122.77 (CH), 125.05 (CH), 126.45 (CH), 126.74 (C), 127.86 (CH × 2), 128.34 (CH × 2), 133.60 (C × 2), 134.73 (C), 135.04 (C), 137.20 (C), 137.38 (C), 145.51 (C × 2). MS (ESI): C_21_H_18_ClN_3_O requires *m*/*z* 363.85, found 364.85 [M+H]^+^. Anal. calcd for C_21_H_18_ClN_3_O: C, 69.32; H, 4.99; N, 11.55. Found: C, 69.53; H, 5.00; N, 11.58.

##### (*E*)-*N*′-(1-(3-Chlorophenyl)ethylidene)-2-methyl-1,4-dihydroindeno[1,2-*b*]pyrrole-3-carbohydrazide (**2c**)

The title compound was prepared from **6** and 1-(3-chlorophenyl)ethan-1-one using the general procedure after stirring for 10 h to afford **2c** as a cream solid (0.15 g, 92.5%), mp > 250 °C. IR (cm^−1^): 1656.85 (C=O), 3245.93 (NH), 3096.27 (NH). ^1^H NMR (400 MHz, DMSO-d_6_) *δ*: 2.40 (s, 3H), 2.59 (s, 3H), 3.73 (s, 2H), 7.06 (t, 1H, *J* = 7.6 Hz), 7.26 (t, 1H, *J* = 8.0 Hz), 7.36 (d, 1H, *J* = 7.2 Hz), 7.45–7.47 (m, 3H), 7.77–7.79 (m, 1H), 7.87 (s, 1H), 9.72 (br s, 1H, NH exch. with D_2_O), 11.77 (s, 1H, NH exch. with D_2_O); ^13^C NMR (100 MHz, DMSO-d_6_) *δ*: 13.46 (CH_3_ × 2), 31.17 (CH_2_), 110.45 (C), 115.84 (CH), 122.79 (CH), 124.82 (CH), 125.06 (CH), 125.64 (CH), 126.45 (CH), 126.74 (C), 128.64 (CH), 130.24 (CH), 133.26 (C), 134.76 (C), 135.02 (C), 137.47 (C × 2), 140.51 (C), 145.51 (C × 2). MS (ESI): C_21_H_18_ClN_3_O requires *m*/*z* 363.85, found 364.85 [M+H]^+^. Anal. calcd for C_21_H_18_ClN_3_O: C, 69.32; H, 4.99; N, 11.55. Found: C, 69.46; H, 5.00; N, 11.57.

##### (*E*)-*N*′-(1-(2-Chlorophenyl)ethylidene)-2-methyl-1,4-dihydroindeno[1,2-*b*]pyrrole-3-carbohydrazide (**2d**)

The title compound was prepared from **6** and 1-(2-chlorophenyl)ethan-1-one using the general procedure after stirring for 10 h to afford **2d** as a cream solid (0.07 g, 44.17%), mp > 250 °C. IR (cm^−1^): 1655.33 (C=O), 3244.32 (NH), 3095.94 (NH). ^1^H NMR (400 MHz, DMSO-d_6_) *δ*: 2.38 (s, 3H), 2.57 (s, 3H), 3.71 (s, 2H), 7.05 (t, 1H, *J* = 7.6 Hz), 7.25 (t, 1H, *J* = 7.6 Hz), 7.34 (d, 1H, *J* = 7.6 Hz), 7.42–7.43 (m, 4H), 7.53 (d, 1H, *J* = 8.0 Hz), 9.67 (br s, 1H, NH exch. with D_2_O), 11.76 (s, 1H, NH exch. with D_2_O); ^13^C NMR (100 MHz, DMSO-d_6_) *δ*: 13.47 (CH_3_), 17.68 (CH_3_), 31.34 (CH_2_), 110.21 (C), 115.74 (CH), 122.71 (CH), 124.94 (CH), 126.35 (CH), 126.77 (C), 127.12 (CH), 129.52 (CH), 129.99 (CH), 130.27 (CH), 131.09 (C), 134.62 (C × 2), 134.92 (C), 137.53 (C), 139.20 (C), 145.45 (C × 2). MS (ESI): C_21_H_18_ClN_3_O requires *m*/*z* 363.85, found 364.85 [M+H]^+^. Anal. calcd for C_21_H_18_ClN_3_O: C, 69.32; H, 4.99; N, 11.55. Found: C, 69.11; H, 4.97; N, 11.51.

##### (*E*)-2-Methyl-*N*′-(1-(thiophen-2-yl)ethylidene)-1,4-dihydroindeno[1,2-*b*]pyrrole-3-carbohydrazide (**2i**)

The title compound was prepared from **6** and 1-(thiophen-2-yl)ethan-1-one using the general procedure after stirring for 48 h to afford **2i** as a brown solid (0.04 g, 24%), mp > 250 °C. IR (cm^−1^): 1656.65 (C=O), 3245.38 (NH), 3096.34 (NH). ^1^H NMR (400 MHz, DMSO-d_6_) *δ*: 2.41 (s, 3H), 2.56 (s, 3H), 3.72 (s, 2H), 7.05 (t, 1H, *J* = 7.6 Hz), 7.16–7.17 (m, 1H), 7.25 (t, 1H, *J* = 7.6 Hz), 7.35 (d, 1H, *J* = 7.2 Hz), 7.42–7.46 (m, 1H,), 7.49 (d, 1H, *J* = 2.8 Hz), 7.55 (d, 1H, *J* = 5.2 Hz), 9.63 (br s, 1H, NH exch. with D_2_O), 11.75 (s, 1H, NH exch. with D_2_O); ^13^C NMR (100 MHz, DMSO-d_6_) *δ*: 13.42 (CH_3_), 14.08 (CH_3_), 31.17 (CH_2_), 110.51 (C), 115.80 (CH), 120.56 (CH), 122.76 (CH), 125.05 (CH), 127.38 (CH), 127.45 (CH), 128.26 (CH), 134.67 (C), 135.03 (C), 137.22 (C), 141.21 (C), 143.63 (C), 145.54 (C × 2). MS (ESI): C_19_H_17_N_3_OS requires *m*/*z* 335.43, found 336.43 [M+H]^+^. Anal. calcd for C_19_H_17_N_3_OS: C, 68.04; H, 5.11; N, 12.53. Found: C, 68.24; H, 5.12; N, 12.57.

##### (*E*)-2-Methyl-*N*′-(1-(thiophen-3-yl)ethylidene)-1,4-dihydroindeno[1,2-*b*]pyrrole-3-carbohydrazide (**2j**)

The title compound was prepared from **6** and 1-(thiophen-3-yl)ethan-1-one using the general procedure after stirring for 48 h to afford **2j** as a cream solid (0.08 g, 56.6%), mp > 250 °C. IR (cm^−1^): 1657.85 (C=O), 3246.53 (NH), 3097.88 (NH). ^1^H NMR (400 MHz, DMSO-d_6_) *δ*: 2.38 (s, 3H), 2.57 (s, 3H), 3.72 (s, 2H), 7.05 (t, 1H, *J* = 7.6 Hz), 7.25 (t, 1H, *J* = 7.6 Hz), 7.35 (d, 1H, *J* = 7.6 Hz), 7.45 (d, 1H, *J* = 7.2 Hz), 7.54–7.57 (m, 2H), 7.91 (s, 1H), 9.54 (br s, 1H, NH exch. with D_2_O), 11.74 (s, 1H, NH exch. with D_2_O); ^13^C NMR (100 MHz, DMSO-d_6_) *δ*: 13.44 (CH_3_), 14.24 (CH_3_), 31.15 (CH_2_), 110.58 (C), 115.81 (CH), 122.77 (CH), 125.01 (CH), 125.06 (CH), 125.53 (CH), 126.44 (CH), 126.60 (CH), 134.67 (C), 135.03 (C × 2), 137.19 (C × 2), 141.51 (C), 145.51 (C × 2). MS (ESI): C_19_H_17_N_3_OS requires *m*/*z* 335.43, found 336.43 [M+H]^+^. Anal. calcd for C_19_H_17_N_3_OS: C, 68.04; H, 5.11; N, 12.53. Found: C, 67.90; H, 5.01; N, 12.50.

### 3.2. Biological Methods 

#### 3.2.1. A375 Cell Culture

A375 cells (ATCC, Rockville, MD, USA) were cultured in Dulbecco’s Modified Eagle Medium (DMEM) (Gibco, Fisher Scientific, Waltham, MA, USA). The medium was supplemented with 10% Fetal Bovine Serum (FBS), 5% L-glutamine, and 5% sodium pyruvate. Cells were incubated at 37 °C in a humidified atmosphere containing 5% CO_2_.

#### 3.2.2. Immunocytochemistry

A375 cells (10^4^/well) were seeded 24 h before treatment with the compounds listed in Table 1 at a concentration of 10 μM. Cells were fixed with 4% paraformaldehyde (PFA) 48 h after. Permeabilization was performed through three washes in PBS containing 0.3% Triton X-100 (PBST), followed by blocking in PBST with 5% FBS. The primary antibody β-tubulin (MA5-16308, Invitrogen, Waltham, MA, USA) was diluted 1:400 in blocking solution and incubated overnight. The following day, cells were incubated with Alexa Fluor 488 anti-mouse secondary antibody (Thermo Scientific, Waltham, MA, USA) and DAPI for 1 h.

#### 3.2.3. Confocal Microscopy

Imaging experiments were performed using an Operetta CLS high-content imaging system (PerkinElmer, Hamburg, Germany) and analysed with the Harmony 4.6 software (PerkinElmer). To assess tubulin integrity, A375 cells were imaged at 40× magnification, capturing 25 fields per sample in both biological and technical triplicates. Data analysis was performed using the following workflow: (1) Find Nuclei, (2) Find Cytoplasm (Tubulin+). Cell dimensions were analysed using the following workflow: (1) Find Nuclei, (2) Find Cytoplasm (Tubulin+), (3) Calculate Morphology Properties—Nuclei (Area), (4) Calculate Morphology Properties—Cytoplasm (Area).

#### 3.2.4. Tubulin Polymerization Assay

The tubulin polymerization assay was performed using a kit supplied by Cytoskeleton (BK011P, Cytoskeleton, Denver, CO, USA). This assay is based on the principle that light scattering by microtubules is proportional to the concentration of the microtubule polymer. Polymerization was measured by excitation at 360 nm and emission at 420 nm at 37 °C for 30 min, with fixed acquisitions every 20 s. Each compound was tested at a concentration of 10 μM, while colchicine (3 μM) was used as a control.

### 3.3. Computational Methods

#### Molecular Docking

A molecular docking study was performed to investigate the feasible binding interactions of the synthesized compounds in the colchicine binding site of tubulin.

The tubulin–colchicine crystal structure was obtained from the Protein Data Bank (PDB ID: 4O2B; 2.30 Å of resolution) [25]. Prior to docking, the protein structure was prepared by retaining only the α- and β-tubulin chains, while water molecules, buffers, co-factors, and ligands were removed to simplify the system.

To validate the docking protocol, colchicine was re-docked into its binding pocket. The resulting pose was consistent with the crystallographic binding mode, with an RMSD of 0.305 Å that was calculated using DockRMSD [26].

The geometry optimization of the ligands was carried out using Avogadro 1.1.1 and AutoDock Tools 1.5.7, with default settings for charge and torsion assignment [27,28].

Docking simulations were performed with AutoDock Vina 1.1.2, setting the grid box with dimensions of 20-20-20 Å (x,y,z) [18]. Parameters included an exhaustiveness of 32, generation of up to 10 binding modes, and an energy range of 4 kcal/mol. The best docking poses were selected based on binding affinity and key interactions within the colchicine site.

Molecular visualizations were generated using BIOVIA Discovery Studio Visualizer version 25.1.0.24284 (Dassault Systèmes BIOVIA Corp., San Diego, CA, USA) [19].

Docking simulations were performed on a PC equipped with a 13th Gen Intel^®^ Core™ i7-13700HX processor (Intel Corporation, Santa Clara, CA, USA) (16 cores, 24 threads) and 32 GB of RAM and running Windows 11 Pro.

## 4. Conclusions

We synthesized a small library of two series of novel 16 *N*-acylhydrazones (**1a**–**j**, **2a**–**d**,**i** and **j**) and evaluated both their antiproliferative and cytotoxic activity against the A375 melanoma cell line, selected for this assay, and their ability to inhibit tubulin polymerization. To this end, the compounds were subjected to an immunocytochemistry assay. Compound **2d** was identified as the most active, and the reduction in the number of nuclei and the massive loss of tubulin in the cytoplasm of A375 cells demonstrated its high toxicity.

Based on the best index of the nucleus/cytoplasm ratio, four compounds, **1g**, **1h**, **2c**, and **2d**, were selected to further evaluate their activity in melanoma cell lines. Two more models of metastatic melanoma (SK-MEL-28 and OMM1) were used and treated with the four selected compounds to evaluate the effect on the microtubules and the cytotoxic effect by the reduction in nuclei during the assay. The same compounds, **1g**, **1h**, **2c**, and **2d**, were used to assess the inhibition of tubulin polymerization; all of them elicited interesting activity. Both derivatives **1g** and **1h**, belonging to series **1**, showed a 25% slowdown inhibition of tubulin polymerization, whereas derivatives **2c** and **2d** caused a slowdown equal to 40% and 60%, respectively. These preliminary studies allow for some considerations regarding this novel class.

In series **1a**–**j**, the introduction of two chlorine atoms in the 3,5-positions (**1g**) and 2,6-positions (**1h**) caused a significant increase in activity. Also, in series 2 (**2a**–**d**,**i** and **j**), bearing a methyl group on the imine-portion carbon, the introduction of the chlorine electron-withdrawing group led to compounds endowed with interesting activity, resulting the 3-Cl-substituted (**2c**) and the 2-Cl-substituted (**2d**) derivatives being the most interesting compounds of their series. These results suggest that the main features responsible for the inhibition of tubulin polymerization in both new synthesized series of tricyclic N-acylhydrazones could be the presence of an electron-withdrawing group in the phenyl ring and the methyl group in the imine carbon.

Moreover, the presence of the methyl group on the imine carbon is responsible for an increase in the inhibitory activity against tubulin polymerization, as was observed for compounds **2c** and **2d**, with 40% and 60% inhibition values, respectively, resulting in **2d** being the most active of the whole series. Among these four compounds, derivative **2c** showed the most favourable docking score overall, and it therefore was selected for binding mode analysis within the colchicine binding site. Compound **2c** showed a consistent interaction pattern, as observed via a hydrogen bond with Asn258, a Pi–donor hydrogen bond with Cys101, a Pi–donor hydrogen bond with Cys101, and six Pi–alkyl interactions with Lys352, Ala250, Leu248, Ala316, Leu255, and Cys241 residues.

The structural superimposition of docked **2c** with colchicine showed that derivative **2c** mimics the key interaction of the alkaloid with tubulin by partial overlapping of similar regions.

The superimposition of the compounds highlights the critical role of the interaction with Asn258, since all the four most active derivatives form a hydrogen bond with this residue. Among them, compound **2c**, with its multiple interactions with Cys101 (Pi–donor hydrogen bond), Lys352, Ala250, Leu248, Ala316, Leu255, and Cys241 (Pi–alkyl interactions), results in being the one with the most extensive interaction network, whereas derivative **2d**, responsible for the highest percentage of slowdown in terms of inhibition of tubulin polymerization, formed a lower number of hydrophobic contacts but showed a Pi–cation interaction with Lys254, with this latter also formed by compound **1h**. Finally, compound **1g** displayed an average profile, with Pi–alkyl contacts involving Leu248, Ala250, and Lys352, keeping the Asn258 hydrogen bond.

Based on these preliminary data, we can conclude that these new *N*-acylhydrazones slow down tubulin polymerization, with the molecular modelling results confirming that the most active compounds **1g**, **1h**, **2c**, and **2d** act as tubulin polymerization inhibitors.

Further studies are in progress to assess the impact of further structural modifications, with the aim of gathering new information for the structure-based development of new molecules and to expand the known landscape of structure–activity relationships, aiming at the development of potent and selective molecules targeting the colchicine site of tubulin. Once a more potent and selective hit is identified, further characterization will be performed.

## Data Availability

The original contributions presented in this study are included in the article/Appendix A. Further inquiries can be directed to the corresponding authors.

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
