# Peer review of "Synthesis of Novel Tricyclic N-Acylhydrazones as Tubulin Polymerization Inhibitors"

_ijms, 2025, doi:10.3390/ijms26189212_

Round 1
Reviewer 1 Report
Comments and Suggestions for Authors
I found this manuscript to be interesting and valuable. It presents a well-conceived study combining chemistry, biological evaluation, and molecular modeling. The compounds and their biological profiles are promising, and the research fits well within the scope of IJMS. However, to meet the journal’s standards, several clarifications, corrections, and additions are necessary. These are outlined below.
I particularly encourage the authors to:
-
Provide additional biological data—especially cytotoxicity results on at least one non-cancerous cell line—to assess selectivity and therapeutic potential;
-
Include ¹H and ¹³C NMR spectra for all synthesized compounds in the supplementary material, to allow readers to verify the reported data and ensure structural integrity and compound purity;
-
Re-examine the NMR assignments, particularly the J values, which contain notable inconsistencies;
-
Clarify the origin of ICâ‚…â‚€ values mentioned in the Introduction;
-
Provide complete microwave synthesis conditions to ensure reproducibility.My comments:
-
The caption uses "Mcf7" instead of the correct format "MCF-7," as used consistently throughout the manuscript. Please correct this. Also, the cell line names in Figure 1 do not need to appear in parentheses.
-
The abbreviation "THFan" for anhydrous tetrahydrofuran is non-standard and confusing. Please use "anhydrous THF" or "THF (anhyd.)" to ensure clarity and scientific accuracy.
-
On page 11, the phrase "¹H- and ¹³C-NMR" is written incorrectly—“and” is missing between the two techniques.
-
The experimental section mentions chloroform-d as a solvent, yet none of the reported spectra appear to have been acquired in this solvent. Please clarify whether any NMR spectra were recorded in chloroform-d. If not, remove it from the experimental description.
-
The description of microwave-assisted synthesis using the Biotage® Microwave Initiator Eight 2.5 is too general. Please provide essential reaction parameters such as microwave power (W), temperature (°C), duration (min), and solvent used.
-
The phrase "¹³C-NMR ppm (DMSO-d₆):" is imprecise. A more accurate format would be:
"¹³C NMR (75 MHz, DMSO-d₆) δ:"
This aligns with the ¹H NMR format already used and ensures consistency and clarity throughout the manuscript. Please verify the frequency used (likely 75 MHz) and apply this format uniformly. -
Several coupling constants in the ¹H NMR data appear misassigned. For example, the triplet at δ = 7.25 ppm is listed with J = 4.0 Hz, but the actual peak separation suggests J ≈ 7.6 Hz, which is chemically more plausible. I recommend revisiting each spectrum to verify the accuracy of chemical shift assignments, multiplicities, and coupling constants. I intend to verify these corrections in the next review round.
-
The authors mention specific ICâ‚…â‚€ values for compound I in HeLa and MCF-7 cells but do not clarify whether these results are original or from previous publications. If the data are cited, please include a reference. If they are from the current study, provide the corresponding viability curves.
-
To support claims of selectivity and safety, cytotoxicity should also be evaluated in at least one non-cancerous cell line. I encourage the authors to perform such studies for the most promising compounds (e.g., 1g, 1h, 2c, 2d) and include these results in the revised version. This will greatly strengthen the biological relevance of the work.
-
Although docking results are presented in detail, there is no quantitative discussion linking docking scores with biological outcomes (e.g., tubulin polymerization inhibition). Please consider including such an analysis or commenting on this limitation.
-
I strongly recommend including the complete ¹H and ¹³C NMR spectra for all synthesized compounds in the supplementary material. This will allow readers to validate the reported assignments and assess compound purity—both essential aspects of structural verification in synthetic and medicinal chemistry.
I found some grammatical mistakes, stylistic inconsistencies, and non-standard scientific phrasing that can be improved"
-
Inconsistent cell line formatting (e.g., “Mcf7” instead of “MCF-7” in Figure 1);
-
Incorrect or uncommon word choices (e.g., “synthetized” instead of “synthesized”; “slowdown inhibition” is redundant);
-
“polimerization” should be “polymerization”;
-
“using as internal standard TMS” should be rephrased as “using TMS as the internal standard”;
- “compound selicited” instead of “compound elicited”?
Author Response
We have reported in red all modifications in text
Comment 1: The caption uses "Mcf7" instead of the correct format "MCF-7," as used consistently throughout the manuscript. Please correct this. Also, the cell line names in Figure 1 do not need to appear in parentheses.
Response 1: We have corrected “Mcf-7” in “MCF-7” in the manuscript and in Figure 1, also deleting the parentheses in the latter.
Comment 2: The abbreviation "THFan" for anhydrous tetrahydrofuran is non-standard and confusing. Please use "anhydrous THF" or "THF (anhyd.)" to ensure clarity and scientific accuracy.
Response 2: Done.
Comment 3: On page 11, the phrase "¹H- and ¹³C-NMR" is written incorrectly—“and” is missing between the two techniques.
Response 3: Done. The phrase "¹H- e ¹³C-NMR…" has been changed in “¹H and ¹³C NMR…".
Comment 4: The experimental section mentions chloroform-d as a solvent, yet none of the reported spectra appear to have been acquired in this solvent. Please clarify whether any NMR spectra were recorded in chloroform-d. If not, remove it from the experimental description.
Response 4: Removed.
Comment 5: The description of microwave-assisted synthesis using the Biotage® Microwave Initiator Eight 2.5 is too general. Please provide essential reaction parameters such as microwave power (W), temperature (°C), duration (min), and solvent used.
Response 5: Thank you for your comment. Microwave assisted synthesis refers to intermediate compounds previously described by us, as reported in the manuscript (ref [7]), therefore the paragraph in 4.1.1. General methods (page 15) “Biotage® Microwave Initiator Eight 2.5 in the standard configuration… were monitored by an IR sensor on the outside wall of the reaction.”) has been deleted.
Comment 6: The phrase "¹³C-NMR ppm (DMSO-d₆):" is imprecise. A more accurate format would be: "¹³C NMR (75 MHz, DMSO-d₆) δ:" This aligns with the ¹H NMR format already used and ensures consistency and clarity throughout the manuscript. Please verify the frequency used (likely 75 MHz) and apply this format uniformly.
Response 6: The phrase “¹³C-NMR ppm (DMSO-d₆):” has been changed in “13C NMR (100 MHz, DMSO-d6) ppm:”.
Comment 7: Several coupling constants in the ¹H NMR data appear misassigned. For example, the triplet at δ = 7.25 ppm is listed with J = 4.0 Hz, but the actual peak separation suggests J ≈ 7.6 Hz, which is chemically more plausible. I recommend revisiting each spectrum to verify the accuracy of chemical shift assignments, multiplicities, and coupling constants. I intend to verify these corrections in the next review round.
Response 7: Thank you for your comment. The chemical shift assignments, multiplicities, and coupling constants of all spectra have been revised, as you can verify.
Comment 8: The authors mention specific ICâ‚…â‚€ values for compound I in HeLa and MCF-7 cells but do not clarify whether these results are original or from previous publications. If the data are cited, please include a reference. If they are from the current study, provide the corresponding viability curves.
Response 8: As reported on page 2, lines 2-7 (references [8,9], page 2, line 4) IC50 for compound I were calculated in our previous work (now references [8-10], page 2, line 3 in the revised manuscript). During the revision we have inserted again ref [8] at the end of the last sentence for convenience (page 2, line 8).
Comment 9: To support claims of selectivity and safety, cytotoxicity should also be evaluated in at least one non-cancerous cell line. I encourage the authors to perform such studies for the most promising compounds (e.g., 1g, 1h, 2c, 2d) and include these results in the revised version. This will greatly strengthen the biological relevance of the work.
Response 9: We thank the reviewer for this comment. While we agree with the reviewer on the importance of this experiment, the limited time allocated for revisions made it impossible for us to carry out this test.
Comment 10: Although docking results are presented in detail, there is no quantitative discussion linking docking scores with biological outcomes (e.g., tubulin polymerization inhibition). Please consider including such an analysis or commenting on this limitation.
Response 10: Thank you for your comment. We have commented the point in section 2.3.5.
Comment 11: I strongly recommend including the complete ¹H and ¹³C NMR spectra for all synthesized compounds in the supplementary material. This will allow readers to validate the reported assignments and assess compound purity—both essential aspects of structural verification in synthetic and medicinal chemistry.
Response 11: Thank you for the recommendation. ¹H and ¹³C NMR spectra for all synthesized compounds have been added in the supplementary material, therefore we have added the sentence “(Fig. S3-S18 in Supplementary material) in paragraph 4.1.1. General methods (page 15, line 20).
Reviewer 2 Report
Comments and Suggestions for Authors
The submitted manuscript IJMS-3787280, titled “Synthesis of novel tricyclic N-acylhydrazones as tubulin polymerization inhibitors” by Corona et al., presents the synthesis and biological evaluation of a new series of N-acylhydrazones, with promising preliminary results as tubulin polymerization inhibitors. The authors report the design, synthesis, and evaluation of compounds 1a–j and 2a–j, highlighting four hits (1g, 1h, 2c, and 2d) with notable activity in both immunocytochemistry and in vitro polymerization assays.
The introduction is particularly well-written, with a contextualization of the medical need and chemical rationale, clearly establishing the motivation for the work. The link to prior studies by the group and the justification for exploring the N-acylhydrazone scaffold are well-articulated and appreciated.
Nevertheless, in its current form the manuscript requires major revision, mainly in the areas of data transparency and biological relevance. Below are the detailed comments:
- Although the synthetic procedures are thorough and 1H and 13C NMR data are reported for all final compounds (1a–j and 2a–j) in the experimental section, copies of the actual NMR spectra are not included in the Supporting Information. Please include the full 1H and 13C NMR spectra for all final compounds 1a–j and 2a–j in the Supporting Information.
- The biological evaluation of antiproliferative activity was performed at a single concentration (10 µM) on A375 cells, and results are primarily discussed in terms of cell counts and nucleus/cytoplasm ratios. However, compounds 1g, 1h, 2c, and 2d stand out due to their effects on tubulin and cell morphology and were selected for further testing in tubulin polymerization assays. Despite this, no ICâ‚…â‚€ values are reported for these compounds, which limits the biological impact of the study. To strengthen the pharmacological profile: Please determine and report the ICâ‚…â‚€ values (in µM) for compounds 1g, 1h, 2c, and 2d in the A375 melanoma cell line and compare the ICâ‚…â‚€ values to clinically used tubulin-targeting agents (e.g., colchicine, vinblastine, paclitaxel) as a reference.
Author Response
We have reported in red all modifications in text
Comment 1: Although the synthetic procedures are thorough and 1H and 13C NMR data are reported for all final compounds (1a–j and 2a–j) in the experimental section, copies of the actual NMR spectra are not included in the Supporting Information. Please include the full 1H and 13C NMR spectra for all final compounds 1a–j and 2a–j in the Supporting Information.
Response 1: Thanks for your comment. ¹H and ¹³C NMR spectra for all synthesized compounds have been added in the supplementary material, therefore we have added the sentence “(Fig. S3-S18 in Supplementary material) in paragraph 4.1.1. General methods (page 15, line 20). Moreover, the chemical shift assignments, multiplicities, and coupling constants of all spectra have been revised in the manuscript.
Comment 2: The biological evaluation of antiproliferative activity was performed at a single concentration (10 µM) on A375 cells, and results are primarily discussed in terms of cell counts and nucleus/cytoplasm ratios. However, compounds 1g, 1h, 2c, and 2d stand out due to their effects on tubulin and cell morphology and were selected for further testing in tubulin polymerization assays. Despite this, no ICâ‚…â‚€ values are reported for these compounds, which limits the biological impact of the study. To strengthen the pharmacological profile: Please determine and report the ICâ‚…â‚€ values (in µM) for compounds 1g, 1h, 2c, and 2d in the A375 melanoma cell line and compare the ICâ‚…â‚€ values to clinically used tubulin-targeting agents (e.g., colchicine, vinblastine, paclitaxel) as a reference.
Response 2: This study presents a screening of a new generation of compounds that affect microtubule polymerization. In this work, we focus on the chemical properties and the proof-of-concept activity related to microtubule destabilization. Here we tested these four compounds for their cytotoxic activity and their ability to kill tumour cells. To strengthen our findings, we tested the same concentration (10 µM) of each compound on two additional melanoma cell lines with phenotypes distinct from A375. Specifically, while A375 cells originate from an invasive melanoma, SK-MEL-28 cells derive from a proliferative melanoma, and OMM1 cells originate from uveal melanoma, one of the most aggressive forms of this cancer type.
In a follow-up study, we will evaluate cytoplasmic rearrangements by analysing mitochondrial activity and morphology, as well as cell metabolism, vesicular trafficking, and estimating the EC50 for all these parameters.
Round 2
Reviewer 1 Report
Comments and Suggestions for Authors
My comments are provided in the attached PDF file

Author Response
Comment 1. The notation for the 1H NMR spectrum is correct:
1H NMR (400 MHz, DMSO-d₆) δ:
However, the notation for the 13C spectra requires correction. Currently, it is written as:
13C NMR (100 MHz, DMSO-d₆) ppm:
Please standardize the notation by using the chemical shift symbol δ instead of “ppm.
Response 1: Done
Comment 2. Do not integrate the peak corresponding to residual water (~3.3 ppm), as this is unnecessary and not standard practice. Similarly, integration of the peak at 0 ppm and the DMSO solvent peak is also unnecessary.
Response 2: Done
Comment 3. In their interpretation of the 1H NMR spectrum of compound 1a, the authors reported: 2.54 (s, 3H) – however, no such signal is present in the spectrum. The signal they describe is in fact visible in the spectrum as a smaller peak to the left of the DMSO signal but is not labelled.
Please label this peak. It does not need to be integrated, as it overlaps with the large DMSO peak, but its chemical shift should be indicated and visible.
Response 3: Done
Comment 4. Most integrals in the spectrum have negative values. This presentation is careless and may confuse readers. Please correct the integration and labelling in the 1H NMR spectrum of compound 1a. Also please correct negative integrals values at other places of the manuscript.
Response 4: Thank you for your comment. It has been done for almost all spectra apart from a single peak at 11.745 for compound 1h (Fig. S10). Unfortunately, now it is impossible for us to acquire the spectrum of the compound as the spectrometer is awaiting technical maintenance intervention.
Comment 5. The signals at 7.26 (t, 1H, J = 7.6 Hz), 7.35 (d, 1H, J = 7.2 Hz), 7.42–7.46 (m, 4H) should be integrated separately in the 1H spectrum shown in Figure S3. Currently, they are integrated together, with a total integral of “–16.42”. If we assume that the signal at 7.06 (t, 1H, J= 7.6 Hz) corresponds to one proton with an integral of “–1.35”, then the above-mentioned signals (6 protons in total) should have a combined integral of 1.35 × 6 = 8.1. However, the current value is nearly twice as high, suggesting a higher number of protons. Separate integration of these signals will likely resolve this issue.
A similar signal in the spectrum of compound 1b was described as a doublet.
Response 5: Done
Comment 6. The signal reported as 7.70 (s, 2H) is not a singlet. The peak clearly shows splitting at the top, indicating it is not a singlet. Furthermore, in compound 1a there is no structural feature that would produce a 2H singlet at this chemical shift. Please correct the peak assignment. In addition, the integral for 2 protons should be about 1.35 × 2 = 2.7, not the observed 5.69, which again suggests a greater number of protons. Maybe this is a problem of integration method, please try to integrate this signal more precisely.
Response 6: Done.
Comment 7. In the 1H NMR spectrum of compound 1b shown in Figure S4, many signals have been integrated unnecessarily, likely due to the automatic integration function of the software. Please do not integrate only signals that are important, as this adds unnecessary clutter and can mislead readers.
Response 7: Done
Comment 8. My most serious concern is the suspicion of manipulation of the spectra. In Figure S4, numerous “square shapes” are visible that appear to be covering parts of the spectrum. For example: Figure S4 – spectrum of compound 1b. Such squares appear in the chemical shift region and cover part of the range between 8.331 and 7.713.
(Below in the spectrum, similar squares are visible, covering peaks and integrals that were likely present in that region. In my opinion, this indicates deliberate and substantial intervention by the authors in the experimental data, which unfortunately constitutes poor scientific practice. One of the integral values, 2.32, also appears to have been inserted, as it differs in color shade and is surrounded by a similar square.)
Figure S5 – spectrum of compound 1c. The 1H NMR spectrum of compound 1c shows signs of similar manipulation, with a signal covered between 10.809 and 8.319. (Below, the authors have covered the peak and its corresponding integral for this chemical shift.)
A similar approach was taken with the signal located between 7.032 and 3.718, as well as the one below 2.506.
The spectrum in this region was also altered, most likely to cover peaks that were originally present. However, the integrals for these peaks were left visible, which allows us to infer that the peaks themselves were concealed.
The above examples illustrate instances of author intervention in the spectra; however, these are not the only ones present in the work. I have found similar spectrum modifications in Figure S6 (compound 1d), S8 (compound 1f), S9 (compound 1g), S12 (compound 1j), S14 (compound 2b), S15 (compound 2c), and S17 (compound 2i). Unfortunately, this appears to be substantial and deliberate manipulation by the authors. I recommend that the authors correct this by providing unmodified spectra, reinterpreting them in accordance with my earlier comments, and supplying the raw NMR files so I can verify the authenticity of the presented data.
Response 8: The raw files (reporting internal laboratory code “nr TF”) for all compounds have been inserted in the Supplementary material file_v3
Comment 9. The labels in Figures 4c and 4d are unclear, particularly the label for the third bar, which is difficult to read due to overlapping letters. Please correct this. It would also be advisable to use slightly smaller significance markers so that they do not overlap with the bars, which would improve the overall readability of the figures. A similar issue occurs in Figures 5c and 5d.
Response 9: Done also in Figures 3C and 3D.

Reviewer 2 Report
Comments and Suggestions for Authors
The responses and the new version of the manuscript provided by the authors comply with what was requested in the previous round. Therefore, I support its publication in IJMS.
Author Response
Comment 1. The responses and the new version of the manuscript provided by the authors comply with what was requested in the previous round. Therefore, I support its publication in IJMS.
Response 1: Thank you for your comment and referring
Round 3
Reviewer 1 Report
Comments and Suggestions for Authors
I would like to thank the authors for their collaboration in revising the manuscript and for the changes introduced. I believe that the presentation of the work is now much improved and meets the standards of the excellent journal IJMS. All of my suggested revisions have been incorporated. Finally, I have one minor remark left, which is more of an editorial than a substantive issue, but it is strictly regulated by IUPAC recommendations, so I kindly ask you to apply it.
Specifically, in scientific writing, particularly in NMR spectroscopy, the coupling constant J is written in italics because it represents a physical quantity, not a unit or descriptive text. According to IUPAC and general scientific style conventions, all physical variables and symbols (such as T for temperature, E for energy, δ for chemical shift, J for coupling constant) must be italicized to distinguish them from surrounding words.
Correct: e.g., J = 7.5 Hz
Author Response
Point-by-point Response to reviewer 1 comments – 3nd round
Comment 1. I have one minor remark left, which is more of an editorial than a substantive issue, but it is strictly regulated by IUPAC recommendations, so I kindly ask you to apply it.
Specifically, in scientific writing, particularly in NMR spectroscopy, the coupling constant J is written in italics because it represents a physical quantity, not a unit or descriptive text. According to IUPAC and general scientific style conventions, all physical variables and symbols (such as T for temperature, E for energy, δ for chemical shift, J for coupling constant) must be italicized to distinguish them from surrounding words.
Response 1: Thank you for your comment. We have reported in the manuscript the corrected italic font for the physical quantities in the manuscript.